# Gene Therapy in Combination with Nitrogen Scavenger Pretreatment Corrects Biochemical and Behavioral Abnormalities of Infant Citrullinemia Type 1 Mice

**DOI:** 10.3390/ijms232314940

**Published:** 2022-11-29

**Authors:** Andrea Bazo, Aquilino Lantero, Itsaso Mauleón, Leire Neri, Martin Poms, Johannes Häberle, Ana Ricobaraza, Bernard Bénichou, Jean-Philippe Combal, Gloria Gonzalez-Aseguinolaza, Rafael Aldabe

**Affiliations:** 1Division of Gene Therapy and Regulation of Gene Expression, CIMA, University of Navarra, 31008 Pamplona, Spain; 2Vivet Therapeutics, S.L., 31008 Pamplona, Spain; 3Department of Clinical Chemistry and Biochemistry, University Children’s Hospital Zurich, University of Zurich, 8091 Zurich, Switzerland; 4Division of Metabolism, Children’s Research Centre (CRC), University Children’s Hospital Zurich, 8091 Zurich, Switzerland; 5Vivet Therapeutics, S.A.S., 75008 Paris, France

**Keywords:** gene therapy, rAAV, urea cycle, hyperammonemia, citrullinemia

## Abstract

Citrullinemia type I (CTLN1) is a rare autosomal recessive disorder caused by mutations in the gene encoding argininosuccinate synthetase 1 (ASS1) that catalyzes the third step of the urea cycle. CTLN1 patients suffer from impaired elimination of nitrogen, which leads to neurotoxic levels of circulating ammonia and urea cycle byproducts that may cause severe metabolic encephalopathy, death or irreversible brain damage. Standard of care (SOC) of CTLN1 consists of daily nitrogen-scavenger administration, but patients remain at risk of life-threatening decompensations. We evaluated the therapeutic efficacy of a recombinant adeno-associated viral vector carrying the ASS1 gene under the control of a liver-specific promoter (VTX-804). When administered to three-week-old CTLN1 mice, all the animals receiving VTX-804 in combination with SOC gained body weight normally, presented with a normalization of ammonia and reduction of citrulline levels in circulation, and 100% survived for 7 months. Similar to what has been observed in CTLN1 patients, CTLN1 mice showed several behavioral abnormalities such as anxiety, reduced welfare and impairment of innate behavior. Importantly, all clinical alterations were notably improved after treatment with VTX-804. This study demonstrates the potential of VTX-804 gene therapy for future clinical translation to CTLN1 patients.

## 1. Introduction

Citrullinemia type I (CTLN1) is a rare autosomal recessive genetic disorder that affects approximately 1 in 250,000 people worldwide [1]. This disease is caused by mutations in the argininosuccinate synthetase 1 (*ASS1*) gene, which encodes the ASS enzyme that catalyzes the synthesis of argininosuccinate in the third reaction of the urea cycle [2]. To date, more than 137 different *ASS1* mutations have been identified in CTLN1 patients, causing an impairment of ASS enzyme activity [3] and blockade of the urea cycle, which results in poor processing and elimination of surplus nitrogen in the liver. ASS deficiency leads to the accumulation of neurotoxic ammonia and to an increase of urea cycle byproducts such as citrulline [4,5,6]. Clinical presentation of CTLN1 patients is heterogeneous and includes an acute neonatal form (the most frequent and “classic” form), a milder and more variable phenotype presenting as late-onset form (the “non-classic” form), and a mild or even asymptomatic phenotype [7]. In the acute neonatal form, hyperammonemic encephalopathy appears within hours to days of life. Symptoms typically include lethargy, seizures, vomiting and coma. If the patient is left untreated, death occurs within days. Irreversible severe brain damage can occur within a few hours and survivors remain at risk of severe decompensation for life. Survivors of the neonatal period may present with neurocognitive dysfunction, the degree of which is only partly related to the maximum ammonia concentration reached in circulation. This suggests that additional factors besides hyperammonemia contribute to the cause of intellectual disability [8]. The late-onset form comes with milder symptoms and neurological outcomes [9]. Interestingly, the residual enzymatic ASS activity in patients can partly predict disease severity. Hyperammonemic events and lower cognitive function are more frequent and more severe in patients with 8% or less ASS activity [10]. Hyperammonemia leads to increased cerebral ammonia that has been shown to be neurotoxic and to be associated with brain edema, astrocyte swelling and reduction of glial fibrillary acid protein, impairing neurotransmission and neuronal activity [11,12].

The standard of care (SOC) for CTLN1 patients consists of daily administration of nitrogen-scavengers to reduce the ammonia concentration in plasma and a very restrictive lifelong low-protein and high-calorie diet [13]. The nitrogen scavengers sodium benzoate and/or sodium phenylacetate (or its prodrug sodium or glycerol phenylbutyrate) are the main compounds used in the clinics to circumvent the urea cycle and reduce the hyperammonemia. In addition, administration of arginine helps to improve ammonia excretion through the urea cycle [14]. Thanks to newborn screening programs, early detection of ASS deficiency is now possible in several countries, allowing an early application of SOC often before the manifestation of first symptoms [15], thereby preventing hyperammonemia and promoting a more favorable cognitive outcome. Nowadays, liver transplantation is the only curative treatment that corrects the underlying disorder, and it may improve neurocognitive outcome when it is performed before manifestation of severe brain damage; however, it cannot reverse established neurological dysfunction [16]. 

Given the low availability of donor organs, the risks and the high costs of liver transplantation, together with and the requirement for a lifelong immunosuppression, new therapeutic options for CTLN1 patients are highly desirable. mRNA replacement therapy has shown promising results in animal models of urea cycle disorders, but it requires lifelong regular therapeutic mRNA infusions to obtain a sustained therapeutic benefit [17,18], which may raise safety concerns regarding long term lipid nanoparticle (LNP) injections. On the other hand, adeno-associated viral (AAV) vectors have emerged as an attractive system for liver-directed gene replacement therapy based on their safety and efficacy. AAV vector-based gene therapy has been successfully employed in the treatment of genetic disorders in preclinical studies, as well as in clinical trials for several genetic diseases such as haemophilia [19,20], Leber’s congenital amaurosis [21] and spinal muscular atrophy. Clinical studies carried out so far have included hundreds of patients, and they have generally shown a good safety profile [22] at doses below 1E14 viral genomes per kg (vg/kg). In animal models of urea cycle disorders, AAV-mediated gene transfer has shown promising results ranging from partial to complete restoration of metabolic function, growth and survival [23,24,25,26,27,28,29,30,31]. These encouraging results have been obtained in a phase 1/2 clinical trial in adults with late-onset ornithine transcarbamylase (OTC) deficiency (NCT02991144), and confirmation will come with a phase 3 confirmatory trial and long-term observation.

Inactivation of mouse *Ass1* gene or substitution of mouse ASS1 threonine 389 by isoleucine (T389I) results in a protein with only 5–10% enzymatic activity, similar to the clinical and biochemical characteristics present in CTLN1 patients [32,33]. *Ass1* inactivation in mice leads to neonatal lethality, whereas mice with the hypomorphic mutation T389I (*Ass1^fold^* mice) show high citrulline blood levels, hyperammonemia, impaired growth and short lifespan. Furthermore, they display significant brain abnormalities including defects in neuronal migration, and the generation of nitric oxide is reduced in the brain. Interestingly, *Ass1* gene delivery using AAVs as vectors has shown partial therapeutic effect in CTLN1 mice, rescuing them from neonatal lethality, reducing circulating metabolites and improving growth [34,35]. However, one month after vector administration, hyperammonemia was only partially reduced and limited growth and weight recovery were observed after two months.

In the present study, we assessed the therapeutic effect of a recombinant AAVAnc80 vector [36] expressing human ASS after its administration to 3-week-old *Ass1^fold^* mice treated from birth to weaning with nitrogen-scavenger agents. We observed that a combination of gene therapy with pharmacological pretreatment prevented the death of *Ass1^fold^* mice and normalized their weight gain six months after vector administration. Moreover, gene therapy treatment was able to improve ureagenesis, resulting in a normalization of ammonia levels and a decrease of citrulline in circulation, which in turn led to a significant improvement of several behavioral defects present in *Ass1^fold^* mice. 

## 2. Results

### 2.1. Combination of VTX-804 Administration with Nitrogen Scavenger Pretreatment Normalizes Survival and Weight Gain of Ass1^fold^ Mice

The *Ass1^fold^* mice carry a hypomorphic mutation in the *Ass1* gene, and reproduce several clinical and biochemical characteristics of CTLN1 patients very early in life [31]. Thus, we decided to administer an rAAV vector that expresses human *ASS1* under the control of a liver-specific promoter (VTX-804) to three-week-old mice. In order to mimic the situation of neonatally diagnosed patients [33], some animals received dietary SOC from birth and prior to VTX-804. Additionally, three animal groups were added as a control: a group of *Ass1^fold^* mice was left untreated, a second group received SOC for three weeks but no vector and a group of WT animals of the same genetic background, age, and gender. Animals received intravenously a single dose of VTX-804 (1 × 10^14^ vg/kg), and efficacy of the treatment was assessed analyzing survival and weight gain for six months after vector administration. 

As can be seen in Figure 1, only 18% and 20% of *Ass1^fold^* mice from the untreated group or receiving the SOC alone survived the seven months experimental period, respectively. In contrast, 100% of the animals receiving SOC in combination with VTX-804 survived, as did WT animals. The group of animals treated with gene therapy alone had a slightly lower survival rate (87.5%) than animals treated with the combination (one of eight mice died five months after vector administration). In correlation with these findings, a combination of SOC and VTX-804 normalized weight gain until the end of the experiment. In the group of animals receiving the vector alone, weight gain rate was slightly lower but still significantly improved with respect to untreated *Ass1^fold^* mice. An interesting and relevant result of the study was that, initially, male and female mice were maintained together after weaning, as *Ass1^fold^* mice are not fertile. Surprisingly four *Ass1^fold^* female mice receiving VTX-804 alone or in combination with SOC recovered fertility. These mice were excluded from all the studies, and male and female treated *Ass1^fold^* mice were thereafter separately housed.

### 2.2. VTX-804 Activates Ureagenesis in Ass1^fold^ Mice and Improves CTLN1 Serum Biomarkers

As indicated, the *Ass1^fold^* mice reliably reproduced CTLN1-associated biochemical alterations such as elevation of ammonia and reduction of citrulline in serum. Thus, to further evaluate the therapeutic efficacy of the treatments described in Figure 1, the level of both parameters was evaluated one, three and six months after vector administration. We found that the administration of VTX-804 to *Ass1^fold^* mice resulted in normalization of ammonia concentration at one and three months; however, at six months this therapeutic effect was only maintained in animals that had been administered with VTX-804 and the pharmacological treatment from birth to weaning (Figure 2a–c). Moreover, at three months post treatment citrulline blood levels were lower in *Ass1^fold^* groups treated with VTX-804, but the difference was significant only in the animals receiving the combination in comparison to the untreated control (Figure 2d). However, by the end of the experiment both treatment groups showed a similar reduction of serum citrulline concentration, which was equivalent to 50% of the levels in the surviving untreated *Ass1^fold^* mice (Figure 2e). Accordingly, ureagenesis evaluation in *Ass1^fold^* mice treated with the combination of gene therapy vector and the pharmacological treatment showed a partial recovery three months after the recombinant vector was administered, but the effect was lost by six months (Figure 2f,g). This indicates that, most likely, a second dose of vector is required to maintain the metabolic correction in relation to liver growth and cell division. Taken together, there was a substantial therapeutic effect associated with the administration of VTX-804 and this was significantly improved when animals were pretreated with the SOC as is evident by the temporary normalization of citrulline and ammonia levels and the restoration of ureagenesis.

### 2.3. VTX-804 Transduction Levels Diminish with Time

We assessed VTX-804 transduction levels and *Ass1* transgene expression in the livers of *Ass1^fold^* mice three and six months after vector administration. Interestingly, nitrogen scavenger pretreatment improved rAAV transduction of *Ass1^fold^* livers since *Ass1^fold^* SOC pretreated animals showed higher numbers of viral genomes at three months after vector administration than animals receiving the vector alone. This difference was reduced by six months (Figure 3a). Accordingly, combination of SOC with gene therapy treatment increased the expression of the therapeutic transgene three months after rAAV administration (Figure 3b). We also quantified the percentage of rAAV-ASS1-positive hepatocytes by in situ hybridization. In accordance with the reduction of AAV genomes quantified by qPCR, we also observed a clear reduction in the percentage of transduced hepatocytes from three to six months after vector administration (Appendix A).

To assess if nitrogen scavengers have a direct effect on AAV-mediated liver transduction independent of the liver disease, WT mice were subjected to a daily inoculation of SOC or vehicle from birth until weaning, and then were injected with the VTX-804 vector. As is presented in Appendix A, rAAV genomes and *Ass1* transgene expression were similar in both groups at three months after vector administration, indicating that nitrogen scavenger agents had no direct effect on liver transduction by rAAV and the effect we observed in CTLN1 mice was, indeed, associated with the improvement of the hepatic condition before administering the rAAV.

### 2.4. The Neurological Phenotype Developed by Ass1^fold^ Mice Is Corrected by VTX-804 Treatment

Many CTLN1 patients have intellectual disabilities, including problems in performing their daily routines. In *Ass1^fold^* mice, the behavioral and neurological involvement has been poorly characterized. Initial studies revealed a defect in the cerebellar morphology and cellularity two weeks after birth, which were corrected by treatment with sodium benzoate and arginine [31]. 

Here, we conducted a battery of cognitive, behavioral, and motor tests in surviving two-month-old *Ass1^fold^* mice to define neurological disabilities present in these animals. First, the rotarod test together with an open field test allowed us to conclude that *Ass1^fold^* mice presented no major locomotor activity defects (Appendix A). Moreover, no learning and memory defects could be observed based on a memory test that used the recognition of novel objects (Appendix A). However, *Ass1^fold^* mice displayed several natural behavior defects, such as reduced exploratory behavior as well as marble burying and nest building activities (Appendix A). Finally, *Ass1^fold^* mice also showed some signs associated with anxiety, such as increased stereotypic jumping and stronger tendency to remain close to walls or thigmotaxis (Appendix A). Therefore, these studies confirmed the presence of neurological comorbidities in *Ass1^fold^* mice resembling those in CTLN1 patients.

Next, we evaluated the effect of the different treatment protocols described above on the behavioral abnormalities observed in two-month-old untreated mice. *Ass1^fold^* animals receiving SOC or VTX-804 alone, or the combination of both, as well as WT mice, were subjected to the open field, marble burying and nest building tests one, three and six months after vector administration (Figure 4), which corresponded to two, four and seven-month-old mice, respectively. Interestingly, untreated four-month-old *Ass1^fold^* mice displayed the same behavioral deficits as two-month-old mice, and SOC treatment alone transiently improved thigmotaxis. By seven months, the majority of untreated *Ass1^fold^* mice had died and the few *Ass1^fold^* mice that survived showed no behavioral abnormalities. Therefore, we have some limitations in obtaining robust conclusions about treatment effectiveness at six months after vector administration because *Ass1^fold^* control animals presented a milder phenotype.

Administration of VTX-804 to three-week-old *Ass1^fold^* mice alone or in combination with SOC normalized the distance covered by the animals in the open field test reaching significance at some points and showing a similar tendency in the others, which was similar to that covered by WT litter-mates, and exploratory behavior along the experiment was also restored (Figure 4a–c). Furthermore, VTX-804 administration also eliminated the differences in thigmotaxis observed between saline treated *Ass1^fold^* and WT mice, and results were comparable at one and three months after vector administration, reaching statistical significance one month after vector administration and two months later when in combination with SOC (Figure 4d–f). Last, analysis of jumping stereotypes revealed that *Ass1^fold^* mice treated with VTX-804 alone or in combination with SOC made a reduced number of jumps per minute compared with untreated *Ass1^fold^* mice. The effect was more robust when mice had been pre-treated with SOC for one-month pre-rAAV administration (Figure 4g–i).

The administration of VTX-804 was able to normalize the exploratory behavior of *Ass1^fold^* mice, which a buried similar number of marbles compared to WT littermates at all time points analyzed (Figure 5a–c). SOC pretreatment had a significant beneficial effect one month after AAV administration. However, the difference between untreated *Ass1^fold^* and WT and *Ass1^fold^* VTX-804-treated mice was only significant in two-month-old animals. Moreover, application of VTX-804 normalized complex, species-specific behavior such as nest building, both three and six months after vector administration (Figure 5d–f). Therefore, all abnormal behaviors observed in CTLN1 mice were corrected at some time point by the VTX-804 treatment.

### 2.5. Histological Brain Alterations Developed by Ass1^fold^ Mice

CTLN1 patients usually present cognitive and behavioral problems caused by chronic states of hyperammonemia, but the factors underlying brain dysfunction are not fully understood. It has been observed that astrocytes bear the brunt of ammonia removal in the brain, and hyperammonemia and glutamine accumulations cause changes in astrocytes, such as swelling, which in functional impairment of the cell. This alteration in astrocyte function leads to neuronal damage. Moreover, hyperammonemia also induces the activation of microglia in the cerebellum and brain cortex. Therefore, as *Ass1^fold^* mice showed behavioral abnormalities and suffer a chronic state of hyperammonemia, we started analyzing the presence of astrocytes and microglia in the brain of these mice. 

Glial fibrillary acidic protein (*Gfap*) is the major protein of intermediate filaments in differentiated astrocytes. There are multiple disorders associated with improper *Gfap* regulation, and it has been described that in moderate or chronic hyperammonemia, *Gfap* expression is decreased, thereby influencing the morphology and function of astrocytes. We obtained brains from four- and seven-month-old WT and *Ass1^fold^* mice to evaluate *Gfap* expression by RT-qPCR and immunofluorescence (IF) to determine if chronic hyperammonemia induced any change in its expression.

Analysis of brain samples revealed a reduction of *Gfap* mRNA expression in *Ass1^fold^* mice compared with WT mice of either age (Figure 6a,b). Furthermore, evaluation of the GFAP positive area in brain sections showed that it was reduced in four-month-old *Ass1^fold^* mice in comparison with their WT littermates, an effect that was not observed when these mice were pretreated with nitrogen scavengers. Moreover, three months later, there were no differences between WT and CTLN1 mice. These results suggest that there was a reduction in the number of astrocytes in *Ass1^fold^* mice at four months of age in addition to a sustained *Gfap* downregulation. Administration of VTX-804 slightly increased the number of GFAP positive cells in *Ass1^fold^* mice three months after vector administration (Figure 6c). Accordingly, *Gfap* mRNA expression was normalized three and six months after VTX-804 was inoculated. 

In addition, we decided to evaluate microglia activation by assessing expression of ionized calcium-binding adapter molecule 1 (*Iba1)*, a marker of microglia activation. No differences between WT and *Ass1^fold^* mice were found, either when analyzing the IBA1 positive area in the brain by IF (Appendix A) or when quantifying *Iba1* mRNA expression by qPCR (Appendix A). These results suggest that there was no activation microglia in *Ass1^fold^* mice.

## 3. Discussion

Inherited metabolic disorders of the liver represent a substantial number of childhood diseases, their treatment often being limited to supportive measures, which only reduce the risk of metabolic crises and impairment of the quality of life. Up to now, the only established curative treatment for many of these diseases is orthotopic liver transplantation. However, with two products commercialized, or close to being commercialized, the long-lasting therapeutic effects of liver-directed gene therapy achieved in clinical trials for hemophilia A and B using recombinant AAV vectors identify AAV-mediated gene therapy as a very attractive alternative to transplantation in inherited monogenic liver disorders [37].

Urea cycle disorders (UCDs) are a group of diseases that affect about 1 in 35,000 newborns [38]. UCDs often present already in newborn patients with extremely high levels of ammonia that exert a direct neurotoxic effect in addition to an increase of cerebral volume causing intracranial hypertension that can induce coma and death [39]. UCD patients can control high ammonia levels by reducing protein in their diet and by nitrogen-scavengers and amino acid supplements. However, the only curative treatment is liver transplantation. Interestingly, gene therapy based on rAAV administered to adult mice has proven to improve lifespan, weight gain and some biochemical parameters in several UCD mouse models, namely for deficiencies of OTC [24,25], ARG1 [24], CPS1 [27], NAGS [40], ASA [41,42] and CTLN1 [34,35]. Moreover, a phase 1/2 clinical trial based on rAAV administration to treat late onset OTC in adults has shown significant therapeutic results (NCT02991144, NCT03636438), leading to the prompt initiation of an ongoing phase 3 randomized, double-blind, placebo-controlled cross-over clinical trial (NCT05345171).

Administration of rAAVs to CTLN1 neonates [34] or young mice [35] led to a transient improvement of survival and a partial recovery of weight gain as well as a reduction of ammonia and citrulline concentration in serum—but only one month after vector administration. In the present study, we found that administration of VTX-804 to juvenile mice had a beneficial effect that lasted at least six months, and was significantly improved when it was combined with SOC treatment. This combination allowed the survival of all treated *Ass1^fold^* mice for 27 weeks, and led to a normal weight gain and fertility recovery in both male and female animals. Accordingly, there was also a positive impact on biochemical parameters, with a normalization of serum ammonia concentration one and three months after vector administration. Evaluation of the benefit of treatments in seven-month-old *Ass1^fold^* mice is not feasible, since most untreated *Ass1^fold^* mice die before this age and the few survivors present with milder biochemical and behavioral alterations than at three months. Therefore, six months after vector administration, we could only compare WT and *Ass1^fold^* treated littermates. Measuring the ammonia concentration in blood confirmed the stronger therapeutic effect when VTX-804 was combined with nitrogen scavenger and arginine supplementation before weaning, as the concentration remained similar to WT litter-mates, while there was an increase in *Ass1^fold^* mice treated with the recombinant AAV only.

Treatment of other UCDs in mice with AAV-mediated gene delivery resulted in different degrees of correction. When adult ASL-deficient mice were treated with a rAAV, all mice survived, and ammonia levels were normalized for one year, even though weight recovery was incomplete [31]. Other UCDs sch as arginase1, NAGS and CPSI deficiencies are less responsive to gene therapy and survival of animals treated with recombinant AAVs was only partially recovered [27,28,40]. On the contrary, the phenotype of the OTC deficiency mouse model was corrected when adult animals were treated with rAAVs, showing normal levels of orotic acid in urine lifelong [25]. Furthermore, these studies revealed that a lower therapeutic dose was required for the normalization of hyperammonemia than of plasma amino acids. In line with this finding, in our *Ass1^fold^* mice, we observed a less pronounced effect on serum citrulline concentration reduction in spite of fully normalizing the ammonia concentration. We also noticed that combination of VTX-804 with SOC pretreatment was able to normalize the citrulline levels three months after vector administration; however, this effect seemed to be transient, and was no longer observed three months later. Interestingly, SOC pretreatment increased AAV-transduction and transcription levels, measured as the presence of AAV genomes and *hASS1* mRNA. This could be one explanation for the better therapeutic effect obtained with the combination treatment. Importantly, this positive impact of SOC on transduction was not observed in healthy mice, which suggests that, most likely, the effect observed in *Ass1^fold^* was associated with the improvement of the condition of the host organ. A similar observation was made in mouse Duchenne muscular dystrophy upon AAV-based gene therapy treatment, when an antisense oligonucleotides pre-treatment that allowed transient expression of dystrophin at the sarcolemma of myofibers improved their condition and limited AAV genome loss after vector administration, eventually resulting in a long-term therapeutic effect [29]. In contrast, in our study this positive impact was lost three months later as a consequence of a reduction in the number of transduced hepatocytes, probably due to liver growth, which led, in turn, to diminished ureagenesis and, consequently, an increased serum citrulline concentration. Together, these data indicate that most likely a second dose of the vector is required to maintain the metabolic correction in this early treated animal model. This would require the combination of a treatment to reduce the AAV neutralizing antibodies generated after the first dose, such as preconditioning with an IgG-degrading enzyme treatment [42].

Hyperammonemia in UCDs is associated with the development of CNS pathology, and is one of the best predictors of mortality and severity of neurological deterioration [15]. There are several CNS changes due to hyperammonemia, including a suggested defect in astrocytes, which is similar to the swollen appearance of Alzheimer type II astrocytes. Thus, neuropathologic evaluation of the brains of patients with UCD has focused on astrocytes [43]. The accumulation of ammonia and glutamine causes astrocyte swelling due to increased glutamine synthetase activity, which limits their role in maintaining cellular homeostasis and leaves neurons more vulnerable to increased ammonia levels [43]. Concordantly, in brains of CTLN1 mice, we observed fewer astrocytes based on *Gfap* expression levels and a reduced number GFAP positive cells, as observed in brains of rats with acute hyperammonemia [44] and hepatic encephalopathy in humans [45]. As mentioned above, VTX-804 administration after weaning prevented accumulation of ammonia for up to three months and six months if mice were additionally treated with nitrogen scavenger compounds from birth to weaning. Six months after vector administration, *Gfap* expression in these animals was similar to that observed in WT mice. This indicated that gene therapy could prevent the astrocyte loss seen in brains of CTLN1 mice when there is chronic hyperammonemia. On the contrary, ASL-deficient mice treated with a therapeutic rAAV had a normalized ammonia concentration in serum, but cerebral pathology persisted [41]. These mice did not show any evidence of astrocyte involvement in disease progression. Instead, induction of apoptosis, and an effect on the cerebral citrulline-NO cycle, was described that was associated with cerebral disease. Consequently, unlike CTLN1 mice, ASL mice require correction of hyperammonemia and neuronal ASL-deficient activity to reduce brain pathology. 

UCD patients frequently exhibit learning difficulties and mental retardation, with neurocognitive dysfunction and behavioral impairment being a common long-term outcome that reflects chronic ammonia/glutamine toxicity [46]. Behavioral testing of CTLN1 mice revealed normal motor skills (Rotarod and Open field speed) and learning and memory capacity (NOR). However, several deficits were observed, such as reduced natural behavior (marble burying and nesting), motor stereotypies (hopping) and limited exploratory activity. These behavioral deficits do not reflect the great variety of intellectual and behavioral deficits observed in many CTLN-patients, which include average IQ in the borderline range, delays in the motor and memory domains, adaptive behavior rated in the range of intellectual disability, as well as attention and social problems [47,48,49]. Nevertheless, we have shown the therapeutic potential of rAAV-based gene therapy applied to young individuals, and proven that this treatment can normalize the behavioral deficits in CTLN1 mice for the duration of the study. Unlike what was observed for several biochemical parameters (ammonia and citrulline) and growth, SOC treatment before weaning had no positive effect on behavior correction promoted by VTX-804. Interestingly, gene therapy of ASL-deficient mice also led to the normalization of the serum ammonia concentration; however, behavior was not fully corrected [31]. In ASL-deficiency, the neurological pathology is different compared to other urea cycle defects [50], and includes a decompensation of the citrulline-NO cycle. Unlike ASS, ASL is expressed in the brain, so that not only the liver, but the brain, must be targeted with the gene therapy vector in order to obtain a more robust neurological recovery [41].

Taken together, we have provided strong evidence that the therapeutic effect of a rAAV vector in young *Ass1^fold^* mice is long-term, corrects hyperammonemia, reduces the citrulline concentration, recovers ureagenesis and corrects behavioral deficiencies. Moreover, daily administration of nitrogen scavengers and arginine from birth to weaning before administration of the therapeutic vector further improved the treatment by normalizing weight gain long-term and permitting the survival of the animals. At least partially, this was the consequence of improved hepatic transduction. Therefore, the combination of SOC with rAAV administration to treat ASS-deficiency could represent a new approximation to treat this condition in young individuals and adults that have already received SOC and stabilized their clinical condition.

## 4. Materials and Methods

### 4.1. Cloning and Construction of rAAV Vector

A synthetic wildtype sequence of human *ASS1* cDNA (NCBI Reference Sequence: NM_054012.4) was obtained from ThermoFisher Scientific (Waltham, MA, USA) and was cloned into pAAV-MCS vector (AAV Helper-Free System—Agilent, Santa Clara, CA, USA). To generate the pAAVEAlbAATh*ASS1* plasmid, this was subsequently inserted into pAAVhASS1 plasmid: the albumin enhancer and the alpha-1 antitrypsin (EAlbAAT) promoter [51] and minute virus (MVM) intronic element upstream of the *hASS1* cDNA and downstream of the bovine growth hormone (BGH) polyadenylation signal. Plasmids were characterized by restriction analysis and sequencing.

### 4.2. Production of rAAV Vectors

The rAAV genome carrying AAV2 ITRs was encapsidated in an AAV-Anc80 capsid. The viral particles were produced by polyethyleneimine-mediated co-transfection in HEK-293T cells (ATCC^®^ CRL-3216^™^). HEK-293T were grown in Dulbecco’s modified Eagle’s medium with pyruvate (DMEM 11995073, GIBCO) supplemented with 10% heat-inactivated fetal bovine serum (FBS 10270-106, GIBCO), 1% penicillin and streptomycin (15140-122, GIBCO) in 150 mm plates until nearly confluent. Nearly confluent cells were co-transfected with three different plasmids—the helper/packaging plasmid (pKan-Anc80AAP-2), an adenoviral helper plasmid (pDF6) and the pAAVEAlbAAT*hASS1* plasmid—using linear polyethyleneimine 25 kDa (Polysciences, Warrington, PA). After 72 h, the supernatant was collected and treated with polyethylene glycol solution (PEG8000, 8% *v*/*v* final concentration) for 48–72 h at 4 °C. It was then centrifuged at 1400× *g* for 15 min, the pellet was resuspended in lysis buffer (50 mM Tris-Cl, 150 mM NaCl, 2 mM MgCl_2_, 0.1% Triton X-100) and kept at −80 °C. After three cycles of freezing and thawing, VPs obtained from cell supernatants and lysates were purified by ultracentrifugation at 350,000× *g* for 2.5 h in an iodixanol (Optiprep 415468 ATOM) gradient, and finally they were concentrated using an Amicon Ultra centrifugal device with Ultracel 100K (UFC910008 Millipore) in a total volume of 2 mL. Titration of VP was performed by qPCR using primers complementary to the AAT region: 5′-CCTGTTTGCTCCTCCGATA-3′ and 5′-GTCCGTATTTAAGCAGTGGATCCA-3′. Viral genomes were extracted from DNAase-treated viral particles using the High Pure Viral Nucleic Acid Kit (Roche). The final vector VTX-804 titer obtained was 1 × 10^13^ vg/mL.

### 4.3. Animal Studies

B6Ei.PAss1*^fold/fold^*/GrsrJ heterozygous mice (*Ass1^fold^*; stock number 006449), were obtained from The Jackson Laboratory and were bred in the animal facility at CIMA to generate homozygous *Ass1^fold^* mice. Mice were housed in ventilated cages (maximum six animals per cage) with a 12 h light–dark cycle in a temperature-controlled room, and they received water and food *ad libitum* using a standard chow diet. Mice were genotyped as described by Jackson laboratories (Protocol 19741: End Point Analysis Assay—Ass1<fold>-EP). 

Three-week-old *Ass1^fold^* mice were injected intravenously with VTX-804 diluted in phosphate-buffered saline (PBS) (1 × 10^14^ vg/kg). A group of animals received 20 µL of nitrogen-scavenger agents (sodium phenylbutyrate (250 mg/kg) and sodium benzoate (180 mg/kg)) supplemented with L-Arginine (100 mg/kg) by intraperitoneal (i.p.) administration from birth until weaning and prior to the VTX-804 administration, whereas another group of animals received just the VTX-804. 

At selected time points throughout the study, and at the end of study, blood samples were collected for assessment of biochemical parameters, ammonia and citrulline concentration. For blood collection, mice were anesthetized with isoflurane (IsoVet [469860], Pyramidal Healthcare) and samples were collected from the facial vein and retro-orbital sinus. 

Animals were sacrificed at four and seven months of age (three and six months after vector administration). At necropsy, liver and brain samples were collected for histological analysis and nucleic acid extraction. All animal experiments and procedures were conducted in compliance with ethical regulations for animal testing and the studies were reviewed and approved by the Institutional Ethical Committee /Universidad de Navarra (protocol numbers: 119-15 and 072-17). Every effort was made to minimize the number of animals used and their suffering.

### 4.4. Analysis of Biochemical Parameters in Serum

Serum was separated from whole blood by centrifugation at 15,000× *g* for 10 min at 4 °C. Serum ammonia and transaminases were quantified using a HITACHI C311 analyzer (Roche) following the manufacturer’s instructions. 

The analysis of citrulline serum levels was carried out by OWL Metabolomics SL (One Way Liver, S.L., Parque Tecnológico de Bizkaia, 502 Building, 48160 Derio, Spain) using mass spectrometry coupled to ultra-high performance liquid chromatography as described [51].

### 4.5. Nucleic Acid Extraction and qPCR

Vector genome copies present in liver extracts were determined by qPCR using iQ™ SYBR^®^ Green (BioRad) in a CFX96 Real-Time Detection System (BioRad) with primers specific for the EAlbAAT liver promoter. Mouse *Gapdh* (glyceraldehyde-3-phosphate dehydrogenase) was used as housekeeping gene.

To analyze gene expression, total RNA was isolated from livers using the Maxwell^®^ 16 LEV simplyRNA Tissue Kit (Promega) according to the manufacturer’s instructions and quantified. Extracted RNA was reverse-transcribed into complementary DNA (cDNA) using M-MLV reverse-transcriptase (Invitrogen). Copies of h*ASS1* cDNA were determined by qPCR using GoTaq qPCR Mastermix (Promega) in a CFX96 Real-Time Detection System (BioRad). Mouse histone expression levels were used for normalization. 

Primers used were: human EAlbAAT promoter: 5′-CCCTGTTTGCTCCTCCGATA-3′ and 5′-GTCCGTATTTAAGCAGTGGATCCA-3′; human *ASS1*: 5′-TCGTGTGGCTGAAGGAACAA-3′ and 5′-AAGAGAGGTGCCCAGGAGGTAG-3′. *Gapdh* 5′-TGCACCACCAACTGCTTA-3′ and 5′-GGATGCAGGGATGATGTTC-3′; Histone: 5′-AAAGCCGCTCGCAAGAGTGCG-3′ and 5′- ACTTGCCTCCTGCAAAGCAC-3′, *Gfap*: 5’-GACCAGCTTACGGCCAACAG-3′ and 5’-TCTCCTCCTCCAGCGATTCA-3’, *Iba1*: 5′-GTCCTTGAAGCGAATGCTGG-3′ and 5′-ATAGCTTTCTTGGCTGGGGG-3′

### 4.6. Ureagenesis

The ureagenesis assay in mice was performed in a fasted state, i.e., food withdrawal for about 3 h prior to first blood collection. All animals (WT, *Ass1^fold^* and *Ass1^fold^* administered with VTX-804) were injected with [^15^N]H_4_Cl. All injections were administered i.p. and blood samples (5 μL) were collected in duplicates by tail vein puncture on a sport saver card (PerkinElmer) at three time points: pre-injection, 30 and 90 min after injection of [^15^N]H_4_Cl. Collected samples were stored at −20 °C until analysis. 

The procedure followed the method as described in [52,53]. Isotopic enrichment was reported as the fraction of urea in the sample that became isotopically labelled ([^15^N]urea), and was calculated according to Patterson and colleagues, i.e., 100 × (Rt − R0)/[1 + (Rt − R0)], where Rt and R0 represent the (m/z + 1) to (m/z + 0) isotope ratio of a given time point (Rt) and the pre-injection (R0) samples, respectively.

### 4.7. Behavioural Studies

A behavioural test battery was performed as described previously [54] to assess motor coordination, balance and physical condition (rotarod test), visuospatial memory (novel object recognition memory test), motor activity, anxiety and hyperactivity (open-field test and stereotypies), normal exploratory behaviour (marble burying test) and abnormal behaviour (nest building test).

### 4.8. Immunofluorescence

Rabbit polyclonal anti-*Gfap* antibody (Dako, Z0344) and rabbit polyclonal antibody anti *Iba1* (Wako, #019-741) were incubated with paraffin sections of formaldehyde-fixed brains. For immunofluorescence, the anti-*Gfap* antibody was used at a dilution of 1:1000, the anti-*Iba1* antibody was used at a dilution of 1:4000 and the sections were further processed with donkey anti-rabbit IgG Alexa Fluor 555 (Invitrogen, A31572) or donkey anti-rabbit IgG Alexa Fluor 488 (Invitrogen, R37188)-conjugated antibody (1:200).

### 4.9. Image Acquisition

Brain slides were scanned with a Vectra^®^ Polaris™ Imaging System (Akoya Biosciences). All images were extracted in parallel with ImageJ 1.52p program (NIH, Bethesda, MD) and then area (µm^2^), ratio of area/area tissue (%) and average intensity were quantified in the entire brain section (excluding cerebellum) using Fiji—ImageJ software. 

### 4.10. Statistical Analysis

Data are presented as mean values ± SEM and were statistically analyzed using a one-way ANOVA or the Kruskal-Wallis test, Bonferroni multiple comparison test, two-way ANOVA test for weight analysis, or unpaired *t* test when only two groups were compared, with GraphPad Prism 9.05 software (GraphPad Software Inc., San Diego, CA, USA). *p* < 0.05 was considered significant.

## 5. Conclusions

This study provides strong evidence supporting the combination of VTX-804 gene therapy and SOC for clinical translation to CTLN1 patients based on the therapeutic effect observed in young CTLN1 mice permitting long term survival of all treated animals normalizing weight gain. This effect was based on the hyperammonemia correction, reduction of citrulline concentration, ureagenesis recovery and correction of behavioral deficiencies. Therefore, the combination of SOC with rAAV administration to treat ASS-deficiency could represent a new approximation to treat this condition in young individuals and adults that have already received SOC and stabilized their clinical condition.

## Figures and Tables

**Figure 1 ijms-23-14940-f001:**
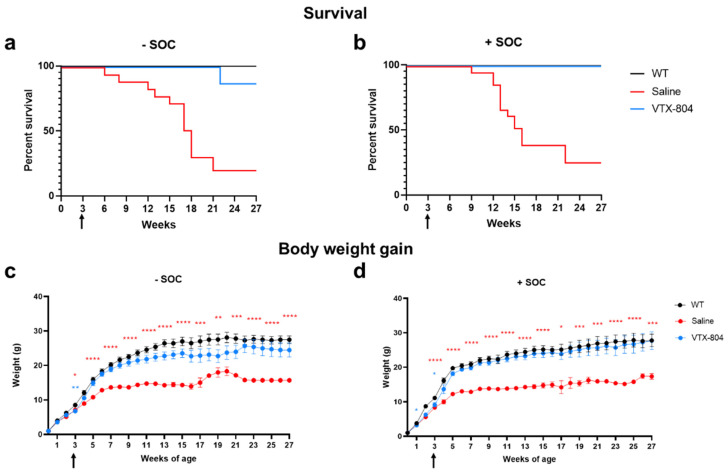
Survival rate and body weight gain of wild-type and *Ass1^fold^* mice. *Ass1^fold^* mice treated with saline or VTX-804 vector with or without pretreatment at three weeks of age and wild-type mice were analyzed for survival (**a**,**b**) and weekly for weight gain (**c**,**d**) for 27 weeks. All three groups were also treated daily with nitrogen scavengers and arginine from birth to weaning (SOC) (**b**,**d**). At three weeks of age, *Ass1^fold^* mice were injected with the VTX-804 vector (n = 16), at a dose of 1 × 10^14^ vg/kg (indicated with an arrow). Controls included saline treated *Ass1^fold^* mice (n = 18) and healthy WT mice (n = 16). * vs. WT mice. Data are mean ± SEM. Two-way ANOVA. * *p* < 0.05, ** *p* < 0.01, *** *p* < 0.001, **** *p* < 0.0001.

**Figure 2 ijms-23-14940-f002:**
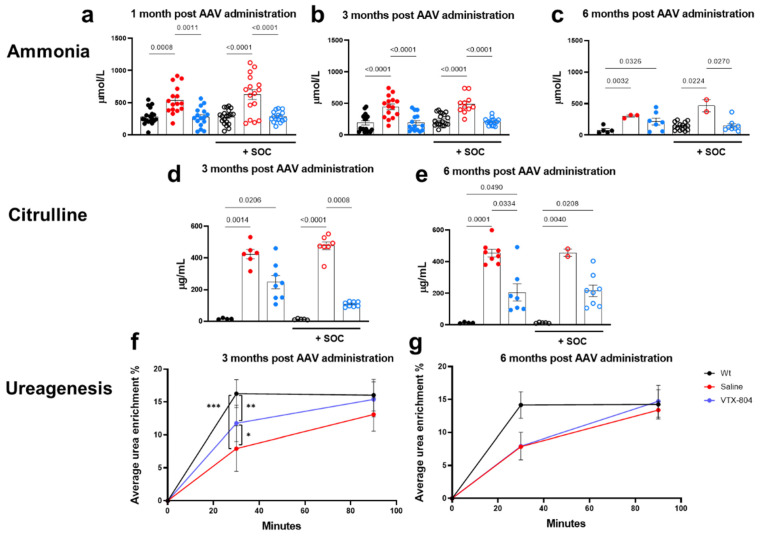
Serum ammonia and citrulline concentration and ureagenesis in wild-type and *Ass1^fold^* mice. *Ass1^fold^* mice treated with saline or VTX-804 vector at three weeks of age and wild-type mice were bled one, three and six months later to quantify ammonia (**a**–**c**) and citrulline (**d**,**e**) concentration. All three groups were also treated daily with nitrogen scavengers and arginine from birth to weaning (SOC) and SOC treated mice were used to assess urea production (ureagenesis) three and six months after vector administration (**f**,**g**). Ammonia groups of animals without SOC: WT mice (black closed circles; n = 19), saline treated *Ass1^fold^* mice (red closed circles; n = 16), *Ass1^fold^* treated with VTX-804 vector (blue closed circles; n = 16). Ammonia groups of animals with SOC: WT mice (black open circles; n = 19), saline treated *Ass1^fold^* mice (red open circles; n = 16), *Ass1^fold^* treated with VTX-804 vector (blue open circles; n = 16). Half of the mice were sacrificed 3 months after vector administration and consequently the six months Wt and VTX-804 groups were composed of less animals. Citrulline groups of animals without SOC: WT mice (black open circles; n = 4), saline treated *Ass1^fold^* mice (red open circles; n = 6 [3 months p.i.] and n = 8 [6 months p.i.]), *Ass1^fold^* treated with VTX-804 vector (blue open circles; n = 8 [3 months p.i.] and n = 7 [6 months p.i.]). Ureagenesis groups of animals: WT mice (black open circles; n = 9 [3 months p.i.] and n = 8 [6 months p.i.]), saline treated *Ass1^fold^* mice (red open circles; n = 10 [3 months p.i.] and n = 2 [6 months p.i.]), *Ass1^fold^* treated with VTX-804 vector (blue open circles; n = 8). WT mice (black circles; n = 4), saline treated *Ass1^fold^* mice (red circles; n = 6 [3 months p.i.] and n = 8 [6 months p.i.]), *Ass1^fold^* treated with VTX-804 vector (blue circles; n = 8 [3 months p.i.] and n = 7 [6 months p.i.]). Only *p* values < 0.05 are shown (one-way and two-way ANOVA with Bonferroni multiple comparisons test or Kruskal-Wallis with uncorrected Dunn’s test), * *p* < 0.05, ** *p* < 0.01, *** *p* < 0.001.

**Figure 3 ijms-23-14940-f003:**
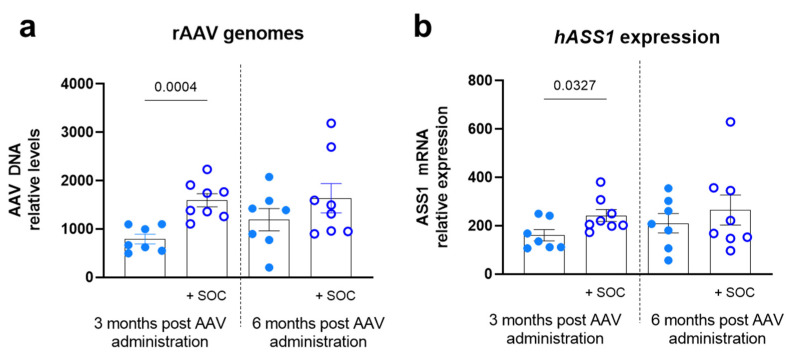
Analysis of liver transduction and transgene expression. Three-week-old *Ass1^fold^* mice were injected with the VTX-804 vector (n = 16), at a dose of 1 × 10^14^ vg/kg in combination or not with pharmacological treatment. Three or six months later, mice were sacrificed for quantification of viral genomes (**a**) and transgene expression (**b**) by qPCR and qRT-PCR, respectively. Total DNA values were normalized against *Gapdh* levels (*10E05) and total mRNA levels against histone (*100). Data were represented as mean ± SEM. Only *p* < 0.05 values are presented (Student *t* test).

**Figure 4 ijms-23-14940-f004:**
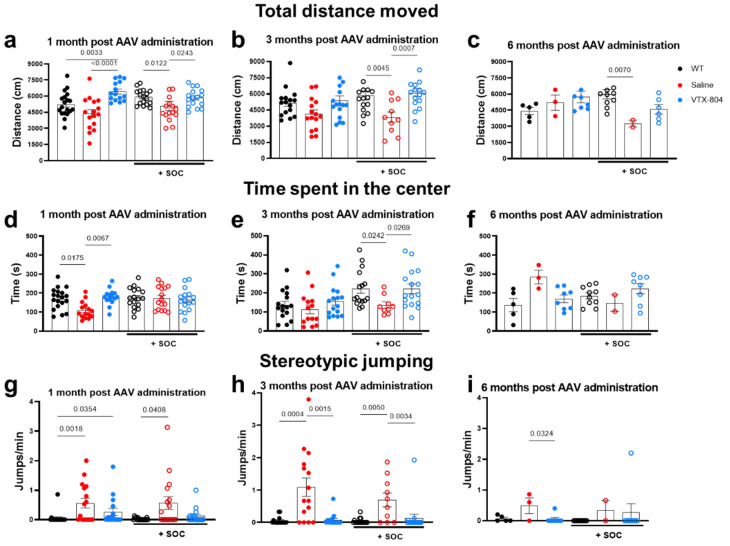
Open field test. *Ass1^fold^* mice treated with VTX-804 vector with or without nitrogen scavenger treatment from birth to weaning (SOC) were subjected to the open field test one, three and six months after vector administration. The following parameters were analyzed: total distance moved (**a**–**c**) and thigmotaxis to quantify the time spent at the center of arena (**d**–**f**). In addition, the number of hopping stereotypes (jumps/min) were quantified at the same time points (**g**–**i**). Group of animals without SOC treatment: *Ass1^fold^* treated with VTX-804 vector only (blue closed circles; n = 16), saline treated *Ass1^fold^* mice (red closed circles; n = 16) and healthy WT mice (black closed circles; n = 19). Group of animals with SOC treatment: *Ass1^fold^* treated with VTX-804 vector (blue open circles; n = 16). Controls included saline treated *ASS1^fold^* mice (red open circles; n = 16) and healthy WT mice (black open circles; n = 19). Some groups were composed of a different number of animals at different measurements due to technical inconsistencies when they were studied and, in the saline group, as consequence of deaths. Half of mice were sacrificed 3 months after vector administration and consequently the six months Wt and VTX-804 groups comprised less animals. Only *p* < 0.05 values are presented (one-way ANOVA with Bonferroni multiple comparisons test or Kruskal-Wallis with uncorrected Dunn’s test).

**Figure 5 ijms-23-14940-f005:**
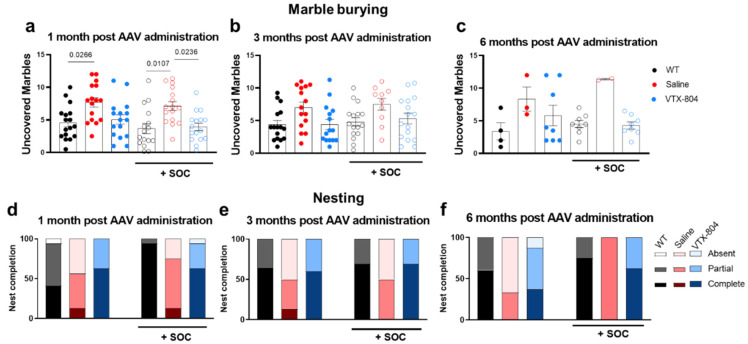
Marble burying and nest building tests. *Ass1^fold^* mice treated with VTX-804 vector with and without SOC from birth to weaning were subjected to the marble burying test one, three and six months after vector administration. Uncovered marbles present in the cage were counted at each time point (**a**–**c**). The nest building capacity was assessed distinguishing completed, partial and absent nest presence (**d**–**f**). Groups of animals without SOC treatment: *Ass1^fold^* treated with VTX-804 vector (blue closed circles; n = 15). Controls included untreated *Ass1^fold^* mice (red closed circles; n = 15) and healthy WT mice (black closed circles; n = 19). Groups receiving SOC treatment were: *Ass1^fold^* treated with VTX-804 vector (blue open circles; n = 16). Controls included *Ass1^fold^* mice without vector administration (red open circles; n = 11) and healthy WT mice (black open circles; n = 19). Half of the mice were sacrificed 3 months after vector administration and consequently the six months Wt and VTX-804 groups comprised less animals. Only *p* < 0.05 values are presented (one-way ANOVA with Bonferroni multiple comparisons test or Kruskal-Wallis with uncorrected Dunn’s test).

**Figure 6 ijms-23-14940-f006:**
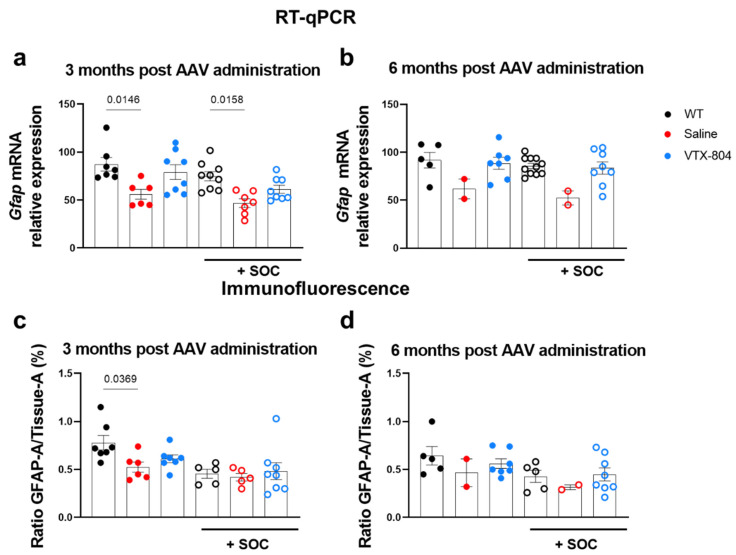
*Ass1^fold^* mice showed a moderate reduction of *Gfap* expression in brain. At 3 weeks of age, *Ass1^fold^* mice were injected with the VTX-804 vector (n = 16), at a dose of 1 × 10^14^ vg/kg in combination or not with pharmacological treatment. Three or six months later, they were sacrificed for quantification of *Gfap* expression by RT-qPCR and normalized against histone (*100) (**a**,**b**) and by immunofluorescence (**c**,**d**). Quantification of GFAP content obtained from automated quantification with Fiji—ImageJ software and ratio of area GFAP/area tissue (%) represented as mean ± SEM. Group of animals without SOC treatment: *Ass1^fold^* treated with VTX-804 vector (blue closed circles; n = 8). Controls included saline treated *Ass1^fold^* mice (red closed circles; n = 6) and healthy WT mice (black closed circles; n = 7). Groups of animals with SOC treatment: *Ass1^fold^* treated with VTX-804 vector (blue open circles; n = 8). Controls included saline treated *Ass1^fold^* mice (red open circles; n = 7) and healthy WT mice (black open circles; n = 9). Only significant differences (*p* < 0.05) are indicated (one-way ANOVA with Bonferroni multiple comparisons test or Kruskal-Wallis with uncorrected Dunn’s test).

## Data Availability

Not applicable.

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
