# Peer review of "Gene Therapy in Combination with Nitrogen Scavenger Pretreatment Corrects Biochemical and Behavioral Abnormalities of Infant Citrullinemia Type 1 Mice"

_ijms, 2022, doi:10.3390/ijms232314940_

Round 1

Reviewer 1 Report

This is an interesting study performed by Bazo and colleagues in which they rescue a mouse model of Citrullinemia I by the combination of scavenger and arginine pretreatment with adenoviral gene transfer. They show the effect of the therapeutic approach on the survival, and analyzed biochemical, behavioral and histological parameters. While the therapy showed clear improvement of biochemical parameters, the behavioral improvement and changes in the expression of Gfap in brain was less pronounced. The study is well designed, but some of the interpretations of the authors would need to be adjusted.

Major points:

1.       The beneficial effect of adenoviral gene transfer on ureagenesis seems to be a tendency, because no significance is indicated (Figure 2f).

2.       It is not clear, why the number of animals is reduced in all groups at the age of six months, although only saline and few vector treated died according to the survival results stated

3.       In the open field test, the effect of vector administration only reaches statistical significance for single time points. This should be made clearer.

4.       The statement “therefore all abnormal behaviors observed in CTLN1 mice are corrected long term by VTX-804 treatment” should be removed

5.       Since study authors suggest the need of a second dose, we would recommend to comment on anti-AAV antibodies (before and) 6 months after vector administration

Minor points:

1.       The numbers of points in the figure does not correspond to the numbers of animals indicated in the figure legends (e.g. group of mice receiving SOC and vector in figure 2a, 3a, 3b)

2.       Number of animals used in the citrulline measurement should be indicated.

3.       In the figure 6 “d” should be labelled “c”, and “e” labelled “d, respectively.

4.       In supplemental figure 4, “KO” should be replace with Ass1fold

5.       In Methods “transgene expression” should read “gene expression”

6.       A description of methods for Iba1 immunofluorescence is missing

Reviewer 2 Report

The authors provide a proof-of-concept study shows the therapeutic effect of rAAV vector for CTLN1 mice. The SOC + rAAV treatment corrects hyperammonemia, reduces citrulline concentration, recovers ureagenesis and corrects the behavioral deficiencies. This study supports the combination of SOC and rAAV treatment for those patients already received SOC. The experiment is well designed, story is clear, and the evidence provided is supportive. I have three major comments. 1. Chandler et al. has reported delivery of hASS1 using AAV8 vector to treat CTLN1 (fold/fold) mice, please demonstrate the novelty of the current study in introduction or discussion to differentiate it from the previous study. 2. The AAV dose used in this study is quite high though it is still clinically relevant. Can the authors provide more information on why use this dose? 3. The rescue effect decreased at 6 months in several assay tested in this study. Recommend having protein level or ASS1 enzymatic activity data includes WT, Saline, and VTX-804 groups. In this way, we can know the initial transduction efficiency and whether the activity of ASS1 is decreased over time.

Minor Comments:

1.     Dose of 1E14 vg/kg is at high end. Have the authors tested the expression of hASS and efficacy of the VTX-804 treatment at a lower dose (1E12 or 1E13)? “Zinn, Eric, et al. "In silico reconstruction of the viral evolutionary lineage yields a potent gene therapy vector." Cell reports 12.6 (2015): 1056-1068.” Showed 3.9E10GC/mouse to 5E11 GC/mouse gave a very high lacZ transgene expression. 

2.     In this study, AAV-Anc80 capsid was used as carrier. Previous study has reported AAV8 is a good carrier to deliver hASS1 gene to treat CTLN1. Will the author able to discuss more about the benefit of AAV-Anc80 over AAV8 or other serotypes? 

3.     Figure 2. f and g, I assume the group are: Wt, Ass1fold + Saline, Ass1fold + VTX-804 + SOC? Have the authors evaluated ureagenesis for VTX-804 alone and SOC alone?

4.     Supplementary figure 2. Please add description (-SOC for blue closed circle, +SOC for purple open circles group). 

5.     Figure 3 and Supplementary figure 2 legend “Three or six months later mice were sacrificed for quantification of viral genomes (a) and transgene expression (b) by qRT-PCR.” can be confusing. Quantification of viral genomes by qPCR, transgene by qRT-PCR

6.     Section 2.3 Authors conclude that VTX-804 transduction and transgene expression levels dimmish with time. From figure 2 a and b, the rAAV genome and hASS1 expression level is smilar or higher at 6 months compared to 3 months when inject alone. The rAAV genome and hASS1 expression level is similar for 3 months and 6 months when combined with SOC. I understand the difference was reduced by 6 months, but the expression level is not decreased. So, the conclusion is misleading. It also rises another question that if the hASS1 expression was not decreased at 6 months, why ureagenesis rescue effect was lost by 6 months. Because loss of enzymatic activity? Recommend having Western blot or enzymatic activity assay. Can authors give some discussion.

7.     Missing Figure 6c

8.     Line 575, Missing conclusions.
